# Traditional Diet and Environmental Contaminants in Coastal Chukotka III: Metals

**DOI:** 10.3390/ijerph16050699

**Published:** 2019-02-27

**Authors:** Alexey A. Dudarev, Valery S. Chupakhin, Sergey V. Vlasov, Sveta Yamin-Pasternak

**Affiliations:** 1Department of Arctic Environmental Health, Northwest Public Health Research Center, 191036 St-Petersburg, Russia; valeriy.chupakhin@gmail.com; 2Northwest branch of Research and Production Association “Typhoon” (RPA “Typhoon”), 199397 St-Petersburg, Russia; 18vsv49@gmail.com; 3Institute of Northern Engineering and Department of Anthropology, University of Alaska Fairbanks, AK 99775, USA; syamin@alaska.edu

**Keywords:** subsistence food, traditional diet, Indigenous people, environmental contaminants, metals, Hg, Pb, As, food safety limits, coastal Chukotka, Russian Arctic

## Abstract

The article is the third in the series of four that present the results of a study on environmental contaminants in coastal Chukotka, which was conducted in the context of a multi-disciplinary investigation of indigenous foodways in the region. The article presents the results of the analysis of metals found in the samples of locally harvested terrestrial, freshwater, and marine biota collected in 2016 in coastal Chukotka. For some species of local fauna and flora, the metals content was demonstrated for the first time. Lead and Hg were low in all foods, while As concentrations were up to four mg/kg ww in fish and marine mammals blubber. Wild plants showed accumulations of Mn (up to 190 mg/kg ww), Al (up to 75 mg/kg ww), Ni, Ba, and Sr. Seaweed contained high levels of As (14 mg/kg) and Sr (310 mg/kg); ascidians (sea squirts) contained Al (up to 560 mg/kg), Cr, and Sr; and blue mussels contained Cd (2.9 mg/kg) and Al (140 mg/kg). Exceedances over the Russian allowable levels were revealed for As, Cd, and Al in different food items. Absence of the established limits for Al and Sr in seafood, and Mn in wild plants and berries, impedes the determination of excess levels. Temporal trends and geographic comparisons of metals in foods have been carried out. The estimated daily intakes (EDIs) of metals by local food consumption were calculated based on the food intake frequencies. Follow-up (15 years after the first study) analyses of Hg, Pb, and Cd concentrations in local foods has not revealed any increase, while a slight decrease tendency was noted for some of the metals in several foods.

## 1. Introduction 

Metals refer to persistent toxic substances (PTS), as they are stable and persistent in the environment, and do not undergo physical, chemical, and bacteriological degradation. The characteristics of some metals include global transportation (also to the Arctic) with atmospheric fluxes, river, sea, and oceanic currents; bioaccumulation and biomagnification in terrestrial and marine food chains; slow elimination from organisms, including humans; and variable toxicity, which can include adverse human health effects. Some metals cross the placental barrier, where they can affect the fetus during its development, and can be excreted in mother’s milk [1]. Some metals are referred by the International Agency for Research on Cancer (IARC) to the agents that are carcinogenic to humans; inorganic arsenic, beryllium and cadmium belong to Group 1 (“sufficient evidence of carcinogenicity”), while inorganic lead, nickel, and cobalt belong to Group 2B (“limited evidence of carcinogenicity”) [2].

Although metals occur naturally in all of the ecosystems and have accumulated and cycled in the environment over geological time periods, significant quantities of metals are now introduced and redistributed in the environment by human activities such as fossil fuel combustion, different industrial processes (e.g., the mining of minerals, metallurgy), agricultural practices, transportation, and waste disposal at local, regional, and global scales. Anthropogenic activities now add metals to surficial environmental compartments at significantly greater rates than natural processes, with the possible exception of volcanic sources [1]. At the global scale, stationary fossil fuel combustion continues to be a major source of heavy metal emissions. Coal combustion contributes chromium (Cr; 69%), mercury (Hg; 66%), manganese (Mn; 85%), antimony (Sb; 47%), selenium (Se; 89%), tin (Sn; 89%), and thallium (Tl; almost 100%) to atmospheric emissions. The combustion of leaded, low-leaded, and ‘unleaded’ gasoline continues to be the major source of atmospheric lead (Pb) emissions [3]. 

Metals are redistributed within the Arctic via atmospheric, marine, freshwater, snow ice permafrost, sediment, and biotic transport mechanisms. The redistribution of metals leads to their presence at concentrations above the naturally occurring levels in locations that are remote from anthropogenic sources, and in forms that are available for biotic uptake. Metals pose a potential risk to the ecosystems and biocenoses of the Arctic [3]. The diet of Indigenous peoples of the Arctic is based mainly on locally harvested foods; therefore, the dietary exposure of the Arctic population to metals remains a health risk problem. 

Metals can accumulate in different organs and tissues of different animals, plants, and fungi. Most metals accumulate in the muscle and parenchymal tissues of fish, birds, and mammals, especially intensively in the liver and kidneys. In the fat of marine mammals, the content of some metals can exceed that of the muscles. The liver of bottom fish species (e.g., freshwater burbot) can accumulate significant levels of certain metals, especially when the water and bottom sediments of the reservoir are highly contaminated by these metals. Seaweed, benthic mollusks (mussels, e.g., bivalves and gastropods), and especially ascidians are able to accumulate extremely high concentrations of metals. The liver and kidneys of reindeer, which feed on lichen “saturated” with metals, also accumulate high levels of metals [1]. Under conditions of pronounced soil contamination, metals can reach high concentrations in both laminar and tubular mushrooms. Mushrooms are powerful “sorbents” of metals from the Arctic soil, including in the territories where there is permafrost, and can be used as indicators of industrial pollution. Wild berries (cranberries, bilberries, cloudberries, etc) and wild plants (wild onions, garlic, leaves and roots of herbaceous and shrub plants), which are often consumed by Indigenous Arctic people, have the ability (in comparison with fungi) to accumulate metals from the soil. Vegetables and garden berries, which are grown in private gardens, sometimes show significant levels of some metals, especially if the gardens are located near the sources of emissions from industrial enterprises [4].

Studies on local food contamination by metals (and dietary human exposure to metals) that have been carried out over the last decades in the countries of circumpolar North have always focused on Hg and occasionally Cd (cadmium) and Pb. The summarized results of these multi-national studies are presented in the Reports of the Arctic Monitoring and Assessment Program (AMAP) [1,5,6,7] and in the Canadian Arctic Contaminants Assessment Reports of the Northern Contaminants Program [8,9].

The only metal that has been comprehensively investigated in different biota species at the circumpolar scale is Hg. A meta-analysis was applied in 83 time-series datasets of Hg in Arctic biota with the purpose of determining whether temporal trends exist in the data collected over the past decades (1975–2008). It was shown that nine significantly increasing time series were found for Canada (30% of the Canadian datasets), two significantly increasing time series were found for Greenland (13%), and one significantly increasing time series was found each for the Faroe Islands (14%) and Iceland (7%). Increasing Hg trends were found in marine invertebrates (one; 8% of the significantly increasing datasets), seabirds (two; 15%), marine mammals (five; 38%) and freshwater fish (five; 38%). Only four (5%) of the time series showed a significantly decreasing Hg trend, and one of these also had a significant non-linear trend component. Two of these were for Atlantic cod from the Faroe Islands and Iceland, one was for Arctic char from Canada, and one was for reindeer from Sweden. None of the Arctic marine mammal or seabird datasets showed a significantly declining trend. The remaining 66 Hg time series showed either no trend or a significant non-linear trend component [7].

The collection and chemical analysis of local foods for Hg, Cd, and Pb were conducted in coastal Chukotka in 2001–2002 along with other Russian Arctic regions within the framework of the Russian Arctic PTS study [10,11]. The community-based dietary and lifestyle survey (interviews of 251 indigenous people in Uelen settlement) based on self-reported daily (weekly, monthly) food frequencies were carried out during the same period. 

The objective of the present part of the project is to carry out an assessment of 18 environmental metals in subsistence species (including fish, terrestrial, and marine mammals, mushrooms, berries, wild plants, and seafood), compare the results of the three metals with those obtained in coastal Chukotka 15 years ago and during the recent follow-up and with other Arctic regions, and estimate the daily intakes (EDIs) of different metals with different food items by the local Indigenous people. The assessment of numerous metals in seafood (seaweed, mussels, and ascidians) has been carried out for the first time in Chukotka coastal waters. Taking into account that residents of the Russian Arctic, including costal Chukotka Indigenous people, eat seafood (including ascidians), the analysis of metals in this group of local foods is of particular interest.

Another important task is the evaluation of metal content (using the same list of metals as for the food analyses) in the drinking water of each of the studied settlements, as the daily intake of metals is formed both by food and by drinking water. Drinking water in the Arctic sometimes is highly contaminated by metals when the settlement is near the ore mining and processing enterprises, e.g., by aluminum in Kirovsk city or by nickel in the Zapolyarny and Nikel cities of the Murmansk oblast [12].

## 2. Materials and Methods

### 2.1. Field Sampling

Collection of samples of local foods and drinking water were carried out in all three study settlements (Enmelen, Nunligran, and Sireniki). Samples of fish (marine, migratory, freshwater), the meat of terrestrial mammals (reindeer, hare), the meat and blubber of marine mammals (whale, walrus, seals), mushrooms, berries, wild plants, seaweeds, ascidians, and mussels have been collected. Most of the samples have been kindly provided by the local people; several samples were collected by the Northwest Public Health Research Center (NWPHRC) specialists during fieldwork (fish by ice fishing, hare by hunting near Enmelen). Birds (except for one goose sample) and the viscera of marine and terrestrial mammals (e.g., liver and kidneys) were not available in the expedition season, and were absent in the storages of local people. 

Samples from multiple specimens of each biological species that were similar in age and size were pooled. Pooled samples accounted for 37% of the total sample number. When pooling the fish species, five to seven specimens were selected, the sizes of which were typical and average for these species; a piece of muscle tissue was cut from the central part of the specimen back (up to the backbone, without affecting it), and packed together with several other pieces that were selected from different specimens of the same species, into one package. Hare meat was taken from the muscles of the lower limbs, while poultry meat was taken from the pectoral muscle, and marine mammal meat (whale, walrus, seal, etc.) was taken from the latissimus dorsi muscle. For the pooling of mushroom samples, seven to 10 caps were used. Wild berries and wild plants were pooled by weight: about 100–150 g. Marine weed and mollusk samples were pooled using five to seven specimens of similar size. After pre-treatment, packing, marking, and freezing, all of the samples were delivered to St. Petersburg in thermocontainers, which helped prevent thawing during transportation. All of the drinking water samples have been collected in each settlement, as well as in Provideniya and Anadyr (for the purpose of conducting the comparison); each 500-mL water sample was put into a plastic container with a screw top. 

The total number of samples that was collected and analyzed for metals is 79 (Table 1); among them were 16 samples of fish, 28 samples of marine mammals, six samples of land mammals, one bird sample, four mushrooms samples, seven berries samples, five wild plants samples, six seafood samples, and six drinking water samples.

### 2.2. Metals Analyzed in Collected Samples

Eighteen metals (Pb, As (arsenic), Cd, Hg, Cu (copper), Zn (zinc), Ni (nickel), Cr, Al (aluminum), Mn, Ba (barium), Sr (strontium), Co (cobalt), V (vanadium), Be (berillium), Mo (molybdenum), Sn, and Sb have been analyzed in all of the samples of local foods (fauna and flora) and in drinking water. 

### 2.3. Chemical Methods and Laboratory Equipment Used for Metals Analysis 

Chemical analyses of metals in all of the samples were performed by the Northwest branch of Research and Production Association "Typhoon" (RPA “Typhoon”), St. Petersburg, Russia, which has international accreditation in the Arctic Monitoring and Assessment Program (AMAP) system. Inductively coupled plasma mass spectrometry (ICP-MS) was applied for the determination of metals, and atomic absorption spectrometry of “cold vapor” (AAS-CV) was applied for the determination of mercury. All of the samples were analyzed on a wet weight (w.w.) basis. 

Before analysis, solid food samples were thawed, dried, homogenized, and digested. An aliquot of a homogenized sample of approximately 1.5 g was weighed. Digestion was carried out by extra pure nitric acid, which was pre-distilled in a DistillAcid apparatus (Berghof, Eningen, Germany), in Teflon tubes in a MARS-5 microwave digester (“CEM” Corp., Matthews, NC, USA). Samples were exposed at 80 °C for 20 min, and then heated to 120 °C for 75 min with pressure control. The mineralized sample was diluted with high-purity water to a volume of 25 mL. Then, 100 mL of drinking water samples were treated by adding five mL of extra pure nitric acid. Analysis of metals in food and water samples was carried out by ICP-MS on a NexION-300D device (“PerkinElmer”, Chicago, IL, USA). Instrument calibration was performed on multicomponent standard solutions ICP-MESS-4 and ICP-MESS-16 (“Certipur”, Merck Chemicals GmbH, Darmstadt, Germany). An additive of yttrium (89Y), which was obtained by dissolving the metal oxide in extra pure nitric acid, was used as the internal standard. Analysis of metals was carried out in the kinetic energy discrimination, Dynamic Reaction Cell™, and normal modes.

An analysis of mercury was carried out without digestion directly by AAS-CV (cold vapor method) on an RA-915M device with a Universal Mercury Prefix console (“Lumeks”, S.Petersburg, Russia). Limit of detection (LOD values) for the analyzed metals in food and drinking water samples were at least 10 times lower than the corresponding established allowable levels. For quality assurance/control purposes, samples for international intercalibrations under the Canadian Northern Contaminants Program quality assurance/control were used. For the certified reference materials, all of the values were within ±20% of the reference values. 

International QA/QC intercalibration standards for metals were used under the aegis of the Canadian Northern Contaminants Program. The obtained results were within ±20% of the reference values.

### 2.4. Processing, Analysis, and Interpretation of the Data

The analyzed concentrations of metals in local foods were evaluated in terms of comparisons between different species in coastal Chukotka and (as much as possible) in the corresponding species in other circumpolar regions, including the assessment of temporal trends of metals. The exceedances over the established limits were determined using Russian food safety regulation standards for metals in raw foods. Estimated daily intakes (EDIs) of metals by local food consumption were calculated based on the average concentrations of metals in local foods and on the intake frequencies of each food item, as reported by the respondents in the questionnaire. Using the hypothetically assumed one portion size as 150 g/meal of each foodstuff, the average annual EDIs of pollutants were calculated for each food group. 

Statistical treatment of the data was carried out using the Microsoft Office 2016 software package.

## 3. Results

### 3.1. Concentrations of Metals in Local Foods

Concentrations of metals (Figure 1) vary highly between food groups. Lead is very low in all foods; it is relatively low in the meat of the terrestrial mammals sampled (up to 2.3 mg/kg ww); As has its highest concentration in fish, marine mammals’ blubber, and seafood; the highest concentrations of Cd are found in seafood, while the highest concentrations of Hg are found in fish and marine mammal meat, the highest concentrations of Cu are found in land mammal meat, and the highest concentrations of Zn are found in the meat of mammals (both marine and terrestrial), and to lesser extent in seafood. Wild plants (particularly *Rhodiola arctica*) accumulate Mn (up to 190 mg/kg ww), Al (up to 75 mg/kg ww), Ni, Ba, and Sr. Seafood (Table 2) is a powerful accumulator of several metals (As, Cd, Cr, Al, Ba, and Sr). Seaweed (*Laminaria saccharina*) contains very high levels of As (14 mg/kg) and Sr (310 mg/kg); ascidians (particularly *Halocynthia aurantium*) are contaminated by Cr, Sr, and Al (up to 560 mg/kg); blue mussels accumulate significant levels of Cd (2.9 mg/kg) and Al (140 mg/kg).

It is particularly important to assess the differences in As content in fish and marine mammal blubber (Figure 2), because these subsistence food groups are the main sources of local people’s dietary exposure to As. It turns out that marine fish and the blubber of bearded seal, walrus, and ice seals are similar in As contamination, while freshwater and migratory fish, as well as whale blubber and mantak, have five to eight times lower concentrations of As.

### 3.2. Follow-Up of Some Metals in the Local Foods in Coastal Chukotka 

Due to the limited number of the collected and analyzed samples, the present Chukotka study cannot provide a reliable or statistically significant comparison with the larger number of samples collected 15 years ago. However, comparison of the levels of Hg, Pb, and Cd was carried out with the data on the same species of fish, reindeer, and marine mammals collected in the neighboring Chukotsky coastal district in 2001–2002 [10]. Table 3 presents the results of this reassessment. Although there was a lack of variation in concentrations of the metals in the majority of the sampled species, there was an obvious tendency for declining metal concentrations in the muscles of arctic char (all three metals), flounder (Pb and Cd), ringed and spotted seals (Hg), and bearded seal (Hg and Pb).

### 3.3. Estimated Daily Intakes of Metals

Figure 3 shows the structure of metals EDIs. While the larger portion of the persistent organic pollutants (POPs) entering the body do so with the consumption of marine mammal blubber, metals are entering the body with other foods. Our investigation shows that seafoods are extremely powerful carriers of metals with the following EDIs: 71–87% of Al, Cd, and Sr; 34% of As and Cr; and 6–11% of Ni, Cu, Zn, and Mn. Fish accounts for a large share of the intake of several metals: 65% of Hg, 43% of As, 21% of Cr, 13–14% of Ni, Cu, and Zn, and also small portions of other metals. Marine mammal meat has a significant share of Cr, Al, Cu, and particularly Hg and Zn (32% and 47%, correspondingly). Marine mammal blubber plays an important role in the intake of As. Land mammals account for half of the Pb intake, 18–19% of Cu and Zn, and 13% of Ni intake. Fowl contributes 50% of Pb and 27% of Cu EDIs. Wild plants are the main contributors of Ni, Mn, and Ba, but they carry also almost all of the metals studied. Berries are a substantial source of Mn, Ni, and Ba intake. It turned out that mushrooms in coastal Chukotka have low metals content. 

### 3.4. Hygienic Regulations of Metals in Raw Foods

The current Russian hygienic regulation states the allowable levels of four metals (Pb, As, Cd, and Hg) in raw foods [13], but covers almost the entire assortment of the products analyzed, including marine mammal meat and blubber, freshwater and marine fish, berries, mushrooms, etc. (Table 4), which is advantageous for food contamination assessment, as the *Codex Alimentarius* [14] has fewer standards. Standards for several other metals (Cu, Zn, Ni, Cr, and Al) present in Table 4 were established in the 1980s for several country foods by the regulations in the former USSR [15,16], and have not been revised. In drinking water (and the water of water bodies), the content of metals (Table 4) is regulated by the existing Russian standards [17].

### 3.5. Exceedances of the Analyzed Concentrations of Metals in Foods Over the Russian Allowable Levels 

The highest concentrations of metals in all of the analyzed samples of mammals, goose, fish, seafood, berries, wild plants, and mushrooms do not exceed the Russian allowable levels for Pb, Hg, Cu, Zn, and Ni. The exceedances for As, Cd, Cr, and Al are presented in Table 5. 

The data of Table 5 show that the majority of exceedances of metals refer to As (in marine mammals blubber, land mammals meat, and seaweed) and Cd (in hare meat, berries, wild plants, and mussels). Certain exceedances over the allowable levels of Cr are observed in berries and wild plants. High exceedances of aluminum are notable in whale meat, wild plants, and bearded seal meat. Unfortunately, due to the absence of the established allowable levels for aluminum and strontium in seaweed, ascidians, and mussels, and manganese in wild plants and berries, there are presently no instruments for evaluating the very high levels of these metals in the given species. 

### 3.6. Drinking Water

Samples of drinking water were collected in three studied settlements (Enmelen, Nunligran, and Sireniki) and also in Provideniya and Anadyr (for comparison). All of the water samples have been analyzed for the same metals as in food. Concentrations of all of the studied metals were much lower than the established hygienic limits, and no exceedances over the Russian allowable levels of metals in water have been detected. The only exception was Cd in Enmelen drinking water (1.5 mcg/L), which is 1.5 times higher than the allowable level for Cd (1.0 mcg/L). The oral TDI of Cd is 60 mcg/person/day; a person drinking an average of two liters of water per day will ingest three mcg/day, which is a negligible addition to the amounts of cadmium entering the organism with Cd-rich seafood, but it is comparable to the consumption of about one kg of fish, land mammal meat, or mushrooms.

## 4. Discussion

### 4.1. Geographic Comparisons

Publications on metals in the fauna and flora in in the Bering Strait region are very limited. The few studies that have been carried out in the far eastern Russian Arctic and in northern Alaska focused on contamination of terrestrial and marine organisms by few metals (mainly Hg, As, and Cd). In the 2011 AMAP Mercury Report [7], which summarized the circumpolar levels and trends of Hg in the biota, no data from Chukotka and Alaska have been presented. In this connection, we include data comparisons with other Arctic regions and older publications. 

#### 4.1.1. Fish

Mercury in sea-run Arctic char was studied in the 1990s in 35 locations in northern Canada. In general, concentrations of mercury were <0.1 mcg/g ww in muscle at all of these locations. Mercury levels in sea-run char from eight communities in Labrador and the Ungava/Hudson Strait region of Nunavik were similar, with means ranging from 0.032–0.040 mcg/g ww. There were no significant differences in Hg levels among the three Labrador sites; however, one location in Nunavik (Quaqtaq) had higher levels than all of the other sites. Mercury levels were slightly lower in sea-run char from Southwest Greenland. Concentrations of Hg in land-locked char sampled in Kangiqsujuaq were about three times higher than in sea-run char. Landlocked populations of Arctic char and Atlantic salmon (*Salmo salar*) from Labrador displayed higher mean Hg values than their oceanic counterparts. Mercury concentrations in the muscle tissue of Arctic cod (*Boreogadus saida*) from the Lancaster Sound region were similar to those observed in Arctic char [9].

A significant increasing trend of Hg have been noted in the period between 1995–2010 in the landlocked char and burbot from the Canadian NWT (up to 0.4 mcg/g ww in muscles). Nevertheless, a decreasing trend of Hg was determined for the same time period in Atlantic cod from Iceland and the Faroe islands [7].

Arsenic was one of the most prominent metals in sea-run Arctic char samples from three communities in Labrador. There were no significant differences in the 1990s in concentrations of As between the three Labrador locations: 0.04–0.06 mcg/g ww. Arsenic levels were significantly higher in sea-run char at two of the four locations in Nunavik compared to the three sites in Labrador [9]. 

Cadmium concentrations are not widely reported for freshwater fish, although existing data show concentrations in muscle to be around <0.005 mg/kg ww. Cadmium in muscle for various species of char from lakes in northern Labrador and Nunavik tended to approach very low detection limits (<0.001 to <0.006 mg/kg ww) [1].

In the present Chukotka study, the concentrations of Hg in fish muscles were (in mg/kg ww): 0.01–0.02 (landlocked char, humpback salmon, and pollack); 0.02–0.03 (cod, saffron cod); 0.04–0.05 (chum, coho, and sockeye salmons); 0.05–0.06 (searun char); and 0.1 (flounder). Concentrations of As were (in mg/kg ww): 0.15–0.2 (landlocked char); 0.25–0.4 (searun char); 0.3–0.45 (chum, coho, and sockeye salmons); 1.3 (flounder); 2.3–2.7 (pollack); and 3.0–4.0 (cod, saffron cod), as shown in Figure 2. Cadmium levels did not differ between all of the freshwater, migrating, or marine species, and were in the range 0.001–0.003 mg/kg ww.

#### 4.1.2. Marine Mammals

Literature on metals in the Arctic marine mammals is rare, and mainly refers to polar bears and belugas, and in a few cases to ringed seals. The analyzed tissues usually include liver and kidneys. The present study is lacking viscera samples of marine mammals, as they were not available during our fieldwork in Chukotka. The marine mammal tissues that have been sampled and analyzed in the present study are meat and blubber, including whale *mantak* (a Yupik name for the whale skin with a thin layer of adjacent blubber, which is called *maktak* in the Inupiaq language) and walrus *kopalkhen* (Chukotkan name of the fermented deboned walrus parts stuffed inside subcutaneous blubber, and aged in subterranean pits, typically over a period of six months).

Only a few publications on metals in marine mammals in Chukotka Alaska are available. Bowhead whale tissues sampled in Barrow, Alaska, in 2002–2003 [18] were analyzed for several metals, part of which we can compare with the present study gray whale that was sampled in Chukotka in 2016. Concentrations of Hg, Cd, Pb, As, Mn, Cu, and Zn in the meat and blubber of Alaskan bowhead whales were very similar to the corresponding concentrations of these metals in Chukotka gray whale meat and blubber. The levels of metals in gray whale *mantak* were relatively similar to the levels in blubber (Table 6).

In the present study, the levels of Hg and Cd in meat and blubber of all marine mammal species were in the range 0.01-0.07 mg/kg ww. Levels of As in the meat of all species were in the range 0.15–0.40 mg/kg ww; while in blubber they decrease from 3.5 to 0.5 mg/kg ww in the sequence walrus—bearded seal—ringed/spotted seal—gray whale (Figure 2). Levels of Pb were very low (<0.05 mg/kg ww) in all samples of marine mammal meat and blubber.

#### 4.1.3. Mussels

Mussels (*Mytilus* spp.) are a particularly common biomonitor in national programs, where ongoing monitoring is conducted at selected locations. Blue mussel (*Mytilus edulis*) data are now available for several Arctic locations, including the Arctic components of national monitoring programs. 

Arsenic was the most prominent of the five metals determined in blue mussels from Nunavik and Labrador. Concentrations ranged from 1.5 mcg/g ww in samples from Makkovik to 2.3 mcg/g ww in samples from Kuujjuaq. Arsenic has been previously found to be present at mcg/g levels in mussels from Nunavik. Mercury levels were low in mussels, ranging from 0.01 to 0.03 mcg/g ww, and did not vary between locations. Organic mercury averaged 54% of the total mercury. This organic mercury is probably in all the methylmercury form. Cadmium levels ranged from 0.2 mcg/g ww to 1.1 mcg/g ww. Similarly low levels of cadmium (0.5 mcg/g ww) and mercury (0.03 mcg/g ww) were found in blue mussels from six communities in the Hudson Bay, Hudson Strait, and Ungava Bay areas. There were no major differences in the levels of metals between mussels collected in Nain and Makkovik and those from Nunavik [9].

Mercury concentrations in blue mussels sampled between 1995–2000 were generally <0.03 mg/kg ww, and no circumpolar trend was apparent. The mean Cd concentrations in blue mussels from different Arctic countries collected between 1995–2000 (there were no samples from the Bering, Chukchi, and Beaufort seas) were in the range 0.2–1.0 mg/kg ww. Mussel concentrations in Qeqertarsuaq, Greenland were high compared to other Arctic locations (1.1 to 2.3 mg/kg ww), which was probably due to the local geological conditions. [1]. 

In the present Chukotka study, Hg levels in all of the sampled invertebrates (including blue mussels) were <0.005 mg/kg ww, while the Cd level in blue mussels was 2.9 mg/kg ww (Table 2).

A Greenlandic monitoring study of metals in blue mussels at the Seqi olivine mine [19] (sampling 2009) showed very similar concentrations of Cu, Zn, Ni, Cr, Co, and V to the corresponding levels in blue mussels sampled in the present Chukotka study; the levels of Pb, As, and Hg were higher; and the levels of Al, Mn, and Ba were three to five times lower (no data on Cd and Sr from Greenland are available). We recalculated the values of dry weight (dw) in the Greenlandic study to wet weight (ww) using the conversion factor of 5.5 for bivalves for comparison with the results of the present study [20]. 

#### 4.1.4. Seaweed

A Greenlandic monitoring study of the metals in brown seaweed *Fucus vesiculosus* at the Seqi olivine mine [19] (sampling 2009) has shown (compared to the corresponding levels in *Laminaria saccharina* in the present study, as indicated in Table 2) that concentrations of Pb, As, Hg, Cu, Cr, Ba, Co, and V were very similar; the levels of Ni and Mn were higher; and the levels of Zn and Al were two to three times lower than in Chukotka’s brown seaweed *Laminaria saccharina* (no data on Cd and Sr from Greenland are available).

The study of metals in *Laminaria saccharina* sampled in 2004 in Kongsfjorden (Svalbard) [21] has demonstrated (compared to the corresponding levels in the present Chukotka study) the similarity of Cd, Ni, and As levels, while concentrations of Zn, Cu, Cr, Mn, and Co were higher in Chukotka (for Zn, Cu, and Cr, it was five to 10 times higher). We recalculated the dw values in both the Greenlandic and Svalbard studies to ww using the conversion factor of 5.1 for Laminaria [20] in order to compare with the results of the present study.

#### 4.1.5. Ascidians

The exceptional water filtration capacity of adult tunicates can sometimes result in the accumulation of pollutants at levels that may be toxic to the tunicate itself, or may make their tissues toxic to predators, including humans. Ascidians may contain metals, including manganese, magnesium, iron, molybdenum, niobium, tantalum, chromium, titanium, and vanadium [22]. Some ascidians species are known for their ability to accumulate certain trace elements from seawater, and are used as indicators of marine pollution in the monitoring of the release of industrial wastes into the marine environment [23,24,25]. In fact, the high resistance of several ascidians to many pollutants explains why they make up such an important part of the fouling fauna in ports all over the world [26]. Farmed tunicates are sometimes grown in bays with coastal pollution, and should be periodically tested to ensure food safety.

Rare metals such as vanadium and cobalt are dissolved in seawater at very low concentrations. Ascidians are known to accumulate extremely high levels of vanadium selectively. Vanadium is accumulated mainly in their blood cells at concentrations corresponding to 10^7^ times higher than that in seawater. The vanadium-binding protein, Vanabin, which is isolated from ascidian, is thought to act as a metal chaperone in ascidian blood cells. Vanabin can be used as an agent for the selective bioaccumulation and biosorption of heavy metals [27]. In the present Chukotka study, the levels of vanadium in ascidians were relatively low (between 0.28–1.7 mg/kg ww), with the highest concentrations in *Halocynthia aurantium*.

Species of ascidians differ in their capacity to accumulate different metals. This may be due to the variations in the availability of metals and their role in the metabolism of a given species. Many researchers have reported variations of metal accumulation in ascidians. A study in the Thoothukudi coast of India showed that among the four metals studied in seawater, Cu accumulated in the highest concentration, followed by Pb, V, and Cd [28]. 

We did not find any literature on the contamination of ascidians by metals in the Arctic, including the areas of the Chukchi, Bering, and Beaufort seas. Therefore, the present study (Table 2) in the communities of the Chukchi Peninsula may be the first to show the contamination levels of ascidians by metals in the coastal waters of the northern Bering Sea. 

## 5. Conclusions

To our knowledge, this is the only study examining multiple metals in the variety of local subsistence foods from the coastal Chukotka since the beginning of the 2000s. For some species of local fauna and flora, the content of metals was demonstrated for the first time.

Levels of metals were very different in various foods. Lead and Hg were low in all of the foods; As levels were up to four mg/kg ww in fish and the blubber of marine mammals. Wild plants (particularly *Rhodiola arctica*) accumulated Mn (up to 190 mg/kg ww), Al (up to 75 mg/kg ww), Ni, Ba, and Sr. Seafood species were highly contaminated by several metals (As, Cd, Cr, Al, Ba, and Sr); seaweed (*Laminaria saccharina*) contained very high levels of As (14 mg/kg) and Sr (310 mg/kg); ascidians (particularly *Halocynthia aurantium*) were contaminated by Cr, Sr, and Al (up to 560 mg/kg); blue mussels accumulated significant levels of Cd (2.9 mg/kg) and Al (140 mg/kg). 

The comparison of metals concentrations in different foods with regard to the food safety limits has revealed the following exceedances over the Russian allowable levels: As in marine mammals blubber, land mammals meat, and seaweed; Cd in hare meat, berries, wild plants, and mussels; and Al in whale meat, wild plants, and bearded seal meat. The absence of the established limits for Al and Sr in seafood, and Mn in wild plants and berries, impedes the determination of the excess levels. Concentrations of all of the studied metals in drinking water samples from all of the studied settlements were much lower than the established hygienic limits. 

The calculated structure of the estimated daily intakes (EDIs) of metals is formed by multiple food items: Pb, mainly by meat of land mammals and fowl; As by seafood and fish; Hg by fish and sea mammal meat; Cd, Al, Sr, and Ba by seafood; and Ni and Mn by wild plants.

Follow-up analysis of Hg, Pb, and Cd in local foods (conducted 15 years after the first study) has not revealed any increase trend; the slight decrease tendency was noted in arctic char (Hg, Cd and Pb), flounder (Pb and Cd), ringed and spotted seals (Hg), and bearded seal (Hg and Pb).

Geographic comparisons based on the available literature (which is scarce for the Bering Strait region) for Hg, As, Cd, and Pb in the main food items, which are monitored at the circumpolar scale, has not revealed new insight.

## Figures and Tables

**Figure 1 ijerph-16-00699-f001:**
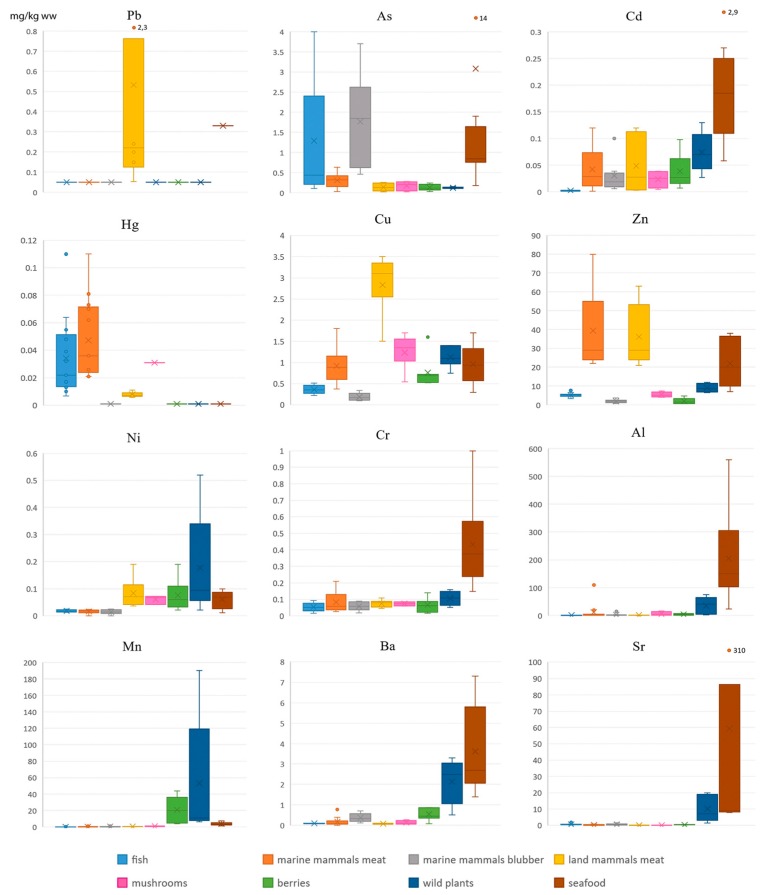
Concentrations of metals in local foods, mg/kg ww.

**Figure 2 ijerph-16-00699-f002:**
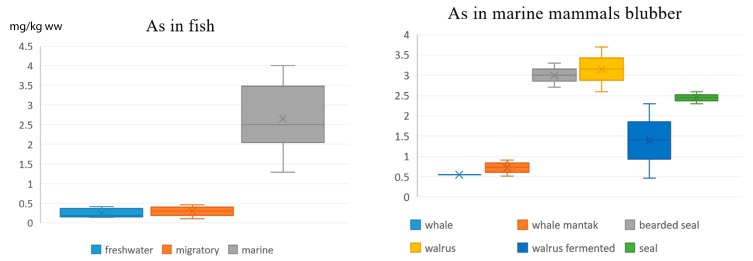
Concentrations of As in fish and marine mammals blubber, mg/kg ww.

**Figure 3 ijerph-16-00699-f003:**
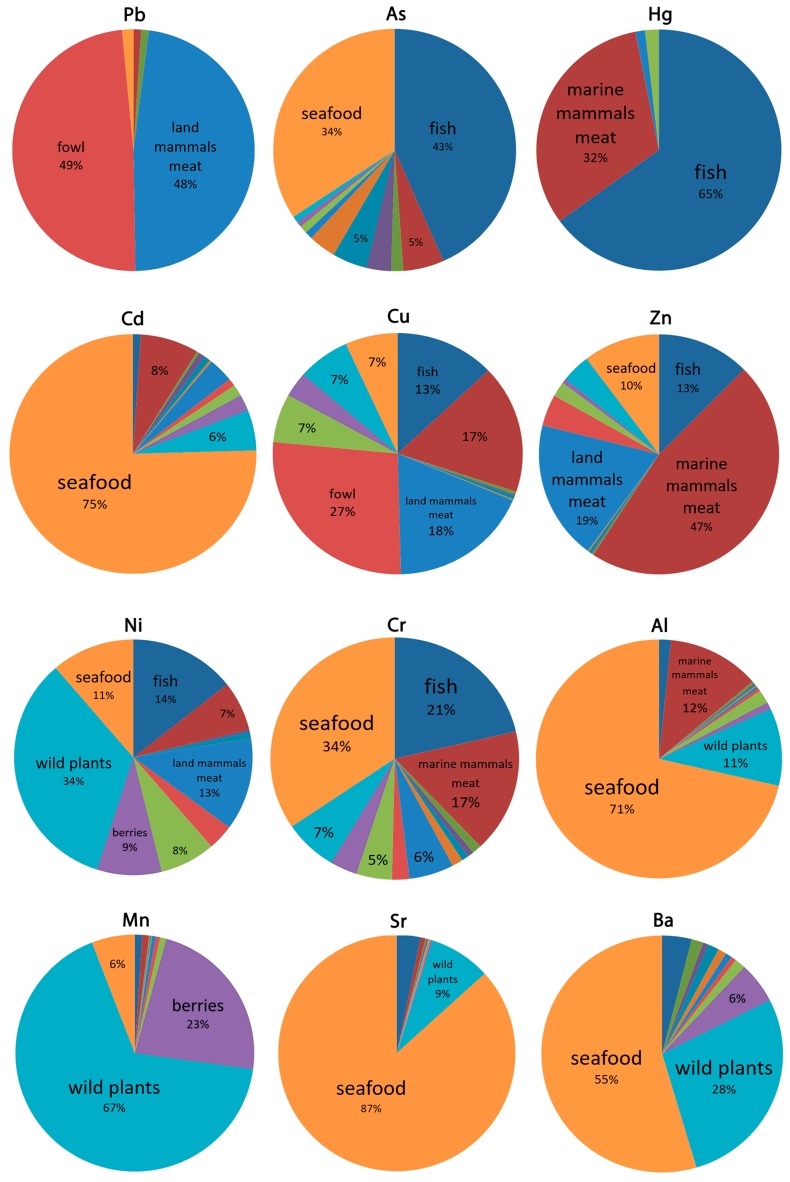
Structure of the estimated daily intakes (EDIs) of metals by local food consumption, %.

**Table 1 ijerph-16-00699-t001:** Collected and analyzed samples of local foods, drinking water, and indoor matters: numbers and locations.

Sample		n Single Samples	n Pools	Location
**Fish**		**11**	**5**	
Arctic char (*Salvelinus alpinus*)		-	1	Achon lake
Arctic char (*Salvelinus alpinus*)		3	-	Enmelen
Grayling (*Thymallus thymallus*)		1	-	Enmelen
Humpback salmon (*Oncorhynchus gorbuscha*)		-	1	Nunligran
Chum salmon (*Oncorhynchus keta*)		1	-	Sireniki
Coho salmon (*Oncorhynchus kisutch*)		1	-	Enmelen
Sockeye salmon (*Oncorhynchus nerka*)		1	-	Nunligran
Pollack (*Pollachius pollachius*)		2	-	Enmelen
Pollack (*Pollachius pollachius*)		1	-	Sireniki
Cod (*Gadus morhua*)		-	1	Sireniki
Starry Flounder (*Platichthys stellatus*)		-	1	Nunligran
Saffron cod (*Eleginus gracilis*)		-	1	Nunligran
Saffron cod (*Eleginus gracilis*)		1	-	Enmelen
**Marine Mammals**		**28**		
Gray whale (*Eschrichtius robustus*)	meat	2	-	Enmelen
	meat	1	-	Nunligran
	blubber	1	-	Nunligran
	mantak	2	-	Enmelen
	mantak	1	-	Nunligran
	mantak	1	-	Sireniki
Pacific walrus (*Odobenus rosmarus)*	meat	2	-	Sireniki
	meat	1	-	Nunligran
	meat	1	-	Enmelen
	blubber	2	-	Enmelen
	kopalkhen	4	-	Enmelen
Bearded seal (*Erignathus barbatus*)	meat	2	-	Sireniki
	meat	1	-	Nunligran
	meat	1	-	Enmelen
	blubber	2	-	Enmelen
Ringed seal (*Phoca hispida*)	meat	1	-	Sireniki
	blubber	1	-	Enmelen
Larga seal (*Phoca largha*)	meat	1	-	Sireniki
	blubber	1	-	Sireniki
**Terrestrial mammals**		**6**		
Reindeer (*Rangifer tarandus*)	meat	2	-	Enmelen
	meat	1	-	Sireniki
Arctic hare (*Lepus arcticus*)	meat	1	-	Enmelen
	meat	1	-	Nunligran
	meat	1	-	Achon lake
**Birds**		**1**		
Snow goose (*Anser caerulescens*)	meat	1	-	Sireniki
**Mushrooms**			**4**	
Yellow-brown cap boletus (*Leccinum testaceo-scabrum*)		-	2	Enmelen
Orange cap boletus (*Leccinum aurantiacum*)		-	1	Enmelen
Orange cap boletus (*Leccinum aurantiacum*)		-	1	Sireniki
**Wild berries**			**7**	
Red bilberry (*Vaccínium vítis-idaéa*)		-	2	Enmelen
Great blueberry (*Vaccínium uliginósum*)		-	1	Enmelen
Crowberry (*Empetrum nigrum*)		-	2	Enmelen
Cloudberry (*Rubus chamaemorus*)		-	2	Enmelen
**Wild plants**			**5**	
Rhodiola Arctic (*Rhodiola arctica*)—leaves		-	2	Enmelen
Wild leek (*Allium sibiricum*)—leaves		-	1	Enmelen
Mixed wild plants—leaves		-	2	Enmelen
**Marine weeds, mollusks, ascidians**		**4**	**2**	
Arctic kelp (*Laminaria Saccharina latissima*)		-	1	Provideniya
Blue mussels (*Mytilus edulis*)		-	1	Enmelen
Sea Squirt (*Dendrodoa aggregata*)		1	-	Enmelen
Sea Squirt (*Boltenia ovifera*)		1	-	Enmelen
Sea peach (*Halocynthia aurantium*)		1	-	Provideniya
Sea peach (*Halocynthia aurantium*)		1	-	
**Drinking water**			**6**	
Lake water		-	1	Enmelen
Lake water		-	1	Nunligran
River water		-	1	Sireniki
Tank water		-	1	Sireniki
Tap home water		-	1	Provideniya
Tap home water		-	1	Anadyr
**Total: 79**		**50**	**29**	

**Table 2 ijerph-16-00699-t002:** The highest concentrations of metals in seafood samples, mg/kg ww.

Metal	Seaweed	Ascidians	Mussels
Pb	<0.05	<0.05	<0.05
As	14	0.87	1.9
Cd	0.18	0.27	2.9
Hg	<0.005	<0.005	<0.005
Cu	0.67	1.7	1.2
Zn	11	38	20
Ni	0.1	0.08	0.07
Cr	0.27	1.0	0.43
Al	23	560	140
Mn	1.2	7.3	3.3
Ba	5.3	7.3	2.3
Sr	310	12	9
Co	0.08	0.2	0.16
V	0.26	1.7	0.3
Be	<0.005	0.02	0.005
Mo	<0.05	<0.05	<0.05
Sn	<0.05	0.08	0.06
Sb	<0.05	<0.05	<0.05

**Table 3 ijerph-16-00699-t003:** Temporal comparisons (2016 with 2001) of metals concentrations in local subsistent foods in coastal Chukotka.

Species	Tissue	Hg	Pb	Cd
arctic char (freshwater)	muscles	↓	↓	↓
flounder (marine)	muscles	→	↓	↓
reindeer	muscles	→	→	→
ringed seal	muscles	↓	→	→
spotted seal	muscles	↓	→	→
bearded seal	muscles	↓	↓	→
walrus	muscles	→	→	→
gray whale	muscles	→	→	→
ringed seal	blubber	→	→	→
spotted seal	blubber	→	→	→
bearded seal	blubber	→	→	→
walrus	blubber	→	→	→
gray whale	blubber	→	→	→

green arrows down = decrease; black arrows flat = no change.

**Table 4 ijerph-16-00699-t004:** Russian allowable levels of metals in raw foods, mg/kg ww, and in drinking water, mg/L.

Food Group	Foodstuff	Pb	As	Cd	Hg	Cu	Zn	Ni	Cr	Al
Land mammals	meat	0.5	0.1	0.05	0.03	5	70	0.5	0.2	10
	liver	0.6	1.0	0.3	0.1	ne	ne	ne	ne	ne
	kidneys	1.0	1.0	1.0	0.2	ne	ne	ne	ne	ne
Marine mammals	meat	1.0	5.0	0.2	0.5	ne	ne	ne	ne	ne
	blubber	1.0	1.0	0.2	0.3	ne	ne	ne	ne	ne
Birds	meat	0.5	0.1	0.05	0.03	ne	ne	ne	ne	ne
	eggs	0.3	0.1	0.01	0.02	ne	ne	ne	ne	ne
Fish	all species	1.0	ne	0.2	ne	10	40	0.5	0.3	30
	freshwater sp.	ne	1.0	ne	0.6	ne	ne	ne	ne	ne
	marine sp.	ne	5.0	ne	0.5	ne	ne	ne	ne	ne
	caviar, milt	1.0	1.0	1.0	0.2	ne	ne	ne	ne	ne
	liver	1.0	ne	0.7	0.5	ne	ne	ne	ne	ne
Seafood	invertebrates	10.0	5.0	2.0	0.2	ne	ne	ne	ne	ne
	seaweed, alga	0.5	5.0	1.0	0.1	ne	ne	ne	ne	ne
Vegetables	all species	0.5	0.2	0.03	0.02	5	10	0.5	0.2	30
Berries	all species	0.4	0.2	0.03	0.02	5	10	0.5	0.1	20
Mushrooms	all species	0.5	0.5	0.1	0.05	10	20	ne	ne	ne
Drinking water		0.01	0.01	0.001	0.0005	1.0	1.0	0.02	0.05	0.2

ne—not established.

**Table 5 ijerph-16-00699-t005:** Exceedances of the highest concentrations of metals in food samples over the Russian allowable levels (% of excess).

Foodstuff	As	Cd	Cr	Al
whale meat	-	-	-	1000% *
walrus blubber	270%	-	ne	ne
fermented walrus blubber	130%	-	ne	ne
bearded seal meat	-	-	-	90%
bearded seal blubber	230%	-	ne	ne
ringed, spotted seal blubber	140%	-	ne	ne
reindeer meat	140%	-	-	-
hare meat	160%	140%	-	-
seaweed	280%	-	ne	ne
mussels	-	45%	ne	ne
berries	-	230%	40%	-
wild plants **	-	330%	60%	275%

ne—not established; * standard for meat of terrestrial mammals; ** standard for berries.

**Table 6 ijerph-16-00699-t006:** Concentrations of selected metals in meat, blubber, and *mantak* samples of gray whale from Chukotka (present study) and bowhead whale from Alaska [18], mg/kg ww (ar mean and range).

Metal	Gray Whale	Bowhead Whale	Gray Whale	Bowhead Whale	Gray Whale
	(Chukotka)	(Alaska)	(Chukotka)	(Alaska)	(Chukotka)
	Meat	Meat	Blubber	Blubber	*Mantak*
Pb	<0.05	nd	<0.05	0.008 (0.006–0.012)	<0.05
As	0.14 (0.03–0.32)	nd	0.55	1.31 (0.77–1.77)	0.72 (0.51–0.91)
Cd	0.01 (0.01–0.02)	0.04 (0.01–0.21)	0.009	0.02 (0.009–0.015)	0.02 (0.006–0.028)
Hg	0.03 (0.02–0.04)	0.02 (0.003–0.04)	<0.005	0.006 (0.005–0.008)	<0.005
Cu	0.45 (0.37–0.55)	0.57 (0.36–0.76)	0.21	0.13 (0.10–0.16)	0.13 (0.11–0.17)
Zn	26.3 (23.2–28.1)	36.3 (24.7–62.8)	1.4	0.93 (0.70–1.16)	1.18 (1.1–1.2)
Mn	0.43 (0.09–1.1)	0.12 (0.05–0.18)	0.14	nd	0.40 (0.098–0.94)

nd—no data.

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
