# Peer review of "Traditional Diet and Environmental Contaminants in Coastal Chukotka III: Metals"

_ijerph, 2019, doi:10.3390/ijerph16050699_

Round 1

Reviewer 1 Report

This document adds useful information concerning tissue concentrations of heavy metals in wild flora and fauna of the arctic, and food contamination information.  Overall, I think you need to be more careful about making certain all of your abbreviations are defined and used consistently, but not at the beginnings of sentences.  Also, I suggest using the word “concentration” rather than “level” when appropriate, since concentration is the preferred and more clear term.  The introduction is long and very poorly referenced, but some of the material is repeated later, so it might be possible to omit those parts.  The introduction should probably mention that some of the metals in the study are required minerals in the diet.  I think the “Discussion” section could be reworked so that you are directly comparing your results to those of previous studies, instead of listing previous study results followed by your results.   You might be able to combined the conclusion with the discussion section.

Some area required clarification in my opinion: 

Why are liver and kidney included in the paper when they were not actually collected and analyzed?  Can this information be omitted, since it doesn’t really add anything to the paper.

The way you estimated daily intake requires further explanation and justification.  It seemed very arbitrary. 

The use of figures was excellent, they were beautiful and attention grabbing, and they really helped me to understand the data you presented.  Make certain that legends are available for the colors and that the colors are clearly different to improve these a little bit and reduce confusion.

Abstract: 

Mention that Chukotka is in the Russian Arctic and home to indigenous people in the abstract.

Spell out the minerals of interest the first time they are used and put the chemical symbols in parentheses (Lead (Pb) and mercury (Hg)).  The mineral name should always be spelled out if it is the first word of a sentence. 

The term “concentrations” is preferred to the word “levels” to improve clarity.

“ascidians (sea squirts) contained Al . . . “

“Follow-up over 15 years of . . . “

Introduction:  The introduction is very long and poorly referenced.  Additionally, some of the information mentioned here is repeated later in the discussion.

First paragraph:

“Metals are persistent toxic substances (PTS) . . . and do not undergo degradation”  Elements can undergo certain chemical changes, but they don’t undergo degradation under normal conditions on earth.

“Characteristics of some metals include global transportation, often with Arctic deposition, accumulation and transport in rivers, seas, and oceans; bioaccumulation and biomagnification, especially in marine food webs; slow elimination from organisms, including humans; and variable toxicity which can include adverse human health effects.”  The metals you tested for are very variable and don’t all have the same characteristics.  Additionally, while one could argue that they can be metabolized to different elemental forms (for example, methylation of metallic mercury by sedimentary microbes) it’s misleading to talk about metabolism of elements, so I changed this to “elimination” which is more clearly definable in this case.  It should also be noted that some of the metals you are looking at are nutrients required for health and belong in the food chain at specific concentrations.

“Some metals cross the placental barrier, where they can affect the fetus during the development, and can be excreted in milk.”

Second paragraph:  “have accumulated and cycled in the environment . . . “

“significant quantitiets of metals are now introduced and redistrubted in the environment from human activities at local, regional, and global scales, including fossil fuel combustion. . . transportation, and waste disposal.

Third paragraph:  references are needed for the first, second, and third sentences.

Fourth paragraph:  This paragraph is also missing references for the information presented.

What does “different intensities” mean in the second sentences? 

 “Seaweed, benthic molluscks (bivalves, eg mussels, and gastropods) . . . “

Note that ascidians are sea squirts here and in the abstract. 

“lichen ‘saturated’ with metals” what do you mean here by “saturated”?  Would it be more clear to say “lichen contaminated with high concentrations of metals?” 

“and can be used as indicators of industrial pollution.”  (omit “of the territories”)

. . . “Natives as foods, also have the ability to accumulate metals from the soil.”

Sixth paragraph:  Why is meta-analysis in quotation marks?  It should say “A meta-analysis was applied in 83 . . .  in Arctic biota with the purpose determining if temporal trends exist in the data collected over the past decades.”  Can you include dates during which the data was collected here?

M&M

Field sampling:  “Samples of fish (marine, migratory, freshwater), meat of terrestrial mammals (reindeer, hare), meat and blubber of marine mammals . . . “

A map of the area of the settlements and other collection areas would be helpful.

Was NWPHRC defined earlier in this manuscript?  You need to spell out the name of this organization once.

Were any bird samples collected besides the goose?  Were any mammalian liver and kidney samples collected?  This is unclear.  If they were not collected, should they be mentioned?

Second paragraph:  A sentence should not start with a number.  “Samples from multiple specimens of each biological species similar in age and size were pooled.  Pooled samples accounted for 37% of the total sample number.”

Spelling error in next sentence: “pieces of approximately equal . . .”  How did you determine that pieces were the same size?  Were they weighed to be certain? 

What size is a “medium sized cap” from a mushroom?  How much does a medium sized cap weigh?

It would be good if you had precise measurements of the samples that were pooled. 

Metals Analyzed section:  This can be combined with the “Chemical Methods” section.

Chemical Methods section:  POPs are persistent organic pollutants, which you are not studying.  Why are they mentioned in the title of this section?

Define RPA.

Define AMAP.

Second paragraph:  I am not familiar with the term “mineralized” being used this way.  We usually use the term “digested” for this process.  Mineralized suggests you are adding mineral to the sample.  Were samples for mercury analysis digested in the same way as samples for the other metals?

Use subscripts appropriately in chemical formulas, or spell out the reagent:  HNO3 = nitric acid.

MARS-5 microwave digester.

LOD needs to be spelled out (limit of detection?)  Also, an abbreviation should not be used as the first word of a sentence.

Processing, analysis:

“Metal concentrations from coastal Chukotka local foods were compared between species, and species from coastal Chukota were compared to the same species in other circumpolar regions to assess temporal trends.” 

Results: 

Figure 1:  “Concentrations of metals. . . “ The blue colors for mushrooms and wild plants are too similar to tell apart.  Additionally, you can’t see the mean and median markers within the bars.

Levels of metals:  Should be “concentrations” of metals.

Sentences should not start with a chemical symbol.  The element name must be spelled out, or the sentence reconstructed: “All foods were very low in Pb”

“Seafood contained the highest concentrations of Cd”

Avoid using dashed lines between the element and information. 

Table 2. “The highest concentrations of metals in seafood samples . . . “

Figure 2:  “Concentrations of As . . . “

Follow-up:  “Although there was a lack of variation in concentrations of the majority of metals, there was an obvious tendency for declining metal concentrations in arctic char . . . “

Table 3:  You need to mention that you are comparing Chukotka and neighboring Chukotsky.

Estimated daily intake:

You already explained what EDI was above, so you can omit the parenthesis.  “Questionnaires were assessed to determine EDIs of metals based on local food consumption and average concentrations of metals . . . “

Does it make sense to estimate 150 g of each foodstuff per meal per day?  I am trying to understand how this worked:  if someone said they consumed whale blubber and mushrooms, you assumed 150 g of each?  This seems both arbitrary and unlikely to be accurate.

Second paragraph:  You mention POPs but I don’t think they were defined before. POP was also used previously in a methods subsection header where it didn’t make sense.

“. . . seafoods are extremely powerful carriers of metals, with the following EDIs:”

Why do you have “makes” in quotation marks after fowl?  “Fowl in the diet contributes 50% of the Pb . . . “

Remove quotation marks from “contributors”

“Berries are a substantial source of Mn. . . “

Figure 3: The pie charts are really helpful and attention-grabbing. You need to have a legend to explain the different colors because some areas of the pie chart are too narrow to read.

Hygienic regulations: 
Allowable levels doesn’t need to be capitalized.  Four should be spelled out.

“which is advantageous for food contamination assessment, as the Codex Alimentarius (13) has fewer standards.”

“Standards for several other metals . . . were established in the 1980s in several country foods by the USSR and have not been revisited. “

Table 4:  Do you need to include liver and kidney if they were not collected in this study?

Exceedances of analyzed concentrations:

Second paragraph:  “Exceedances over the allowable levels of Cr are observed . . . “

Drinking water:  you can use Cd for cadmium throughout this paragraph.

“ingest” doesn’t need to be in quotation marks.

Discussion:  This section needs a little reorganization so that you incorporate your data into the discussion better.  Compare your findings directly to findings from other studies, rather than listing the findings from the other studies and then summarizing your results for those elements at the very end of the section. 

Don’t begin a sentence with an abbreviation, but you can use the chemical symbols for elements elsewhere throughout this section.

Genus and species names should be italicized. 

Table 6:  Concentrations of selected metals in meat, blubber, and mantak sampels of grey whale from Chukotka (present study) and bowhead whale from Alaska . . .”

Conclusions:  Can you say that the study is comprehensive if you’ve already noted that you don’t have enough samples for statistical power?

Second paragraph:

First sentence:  “. . . has revealed the following exceedances over the Russian Allowable Levels: “  The source of the exceedance does not need to be in parenthesis.  You can separate the different groups of elements and foodstuffs using semicolons instead of commas to reduce confusion here.

Author Response

Comments and Suggestions for Authors

·          This document adds useful information concerning tissue concentrations of heavy metals in wild flora and fauna of the arctic, and food contamination information.  Overall, I think you need to be more careful about making certain all of your abbreviations are defined and used consistently, but not at the beginnings of sentences. 

Answer: all the metals symbols have been defined in the text the first time they are used. Also all the names of metals are spelled out if it is the first word of a sentence.

·          Also, I suggest using the word “concentration” rather than “level” when appropriate, since concentration is the preferred and more clear term.

Answer: In our view the words “concentration” and “level” can be used interchangeably in this context, and we choose to use both in order to avoid the constant using the word “concentrations,” which has many instances of use in the text. We have substituted the word “levels” by the word “concentrations” in some places (subsections names, table’s names). We have left the use of “levels” in Russian Allowable Levels, because it is the literal translation of the title.

·          The introduction is long and very poorly referenced, but some of the material is repeated later, so it might be possible to omit those parts.

 Answer: In our view the length of the Introduction is appropriate and effective in introducing the work. We have added references and reorganized the material to some extent.

·          The introduction should probably mention that some of the metals in the study are required minerals in the diet. 

 Answer: Yes, some of the metals in the study could be referred to micronutrients or essential elements (if the “dose” of their ingestion is very small), but micronutrients, as well as the physiological aspects and health effects of exposure to metals, are beyond the scope of the current study.

·          I think the “Discussion” section could be reworked so that you are directly comparing your results to those of previous studies, instead of listing previous study results followed by your results.   You might be able to combine the conclusion with the discussion section.

 Answer: this is exactly what we did in the “Discussion” section – every available data of the previous studies in the Arctic has been compared to our present Chukotka results. The problem is in the lacking of any previous data on many metals. The only metal which has been comprehensively investigated in different biota species at circumpolar scale is Hg. For some species of local fauna and flora the content of metals was demonstrated for the first time.

Some area required clarification in my opinion: 

·          Why are liver and kidney included in the paper when they were not actually collected and analyzed?  Can this information be omitted, since it doesn’t really add anything to the paper?

 Answer: Yes, unfortunately the viscera of marine and terrestrial mammals (e.g. liver and kidneys) were not available at the time of our fieldwork and were absent in the storages of local people. We do have to emphasize that although liver and kidney are generally strong accumulators of some metals and should be regarded as additional sources of total dietary exposure to metals, however the amounts of liver and kidney consumed by local Chukotka people are relatively insignificant.

·          The way you estimated daily intake requires further explanation and justification.  It seemed very arbitrary. 

 Answer:  1) It was impossible within our research setting to administer a personalized 24-hour dietary recall survey. We therefore conducted a broad-based survey on dietary consumption practices, using standard questionnaire. Standard questionnaires are known to be less effective in reflecting the actual consumption practice (particularly in the Arctic), but that was the best available option. 2) We are not nutritionists; we are environmental health researchers. Our main task is to assess the dietary food exposure (to contaminants) of a community, based on the values of local food contamination. 3) The principle to which we adhere over the many years of research in the Russian Arctic is as follows: a thorough assessment of the frequencies of each food item intake is much more informative and reliable as a criterion of real food intakes than an estimation of a portion size; 4) individual fluctuations of the self-reported portion size vary substantially and we need to be able to average these numbers per day/week/month/year; 5) Given that the assumed rage of 100g to 200g is used for meat and fish portions (as well as side dish or salad) around the world, it is easier and more reliable to use conventionally assumed one portion size as 150g/meal of each foodstuff, and then to average the reported frequencies: 1-3 meals/day; 4-6 meals/week; 1-3 meals/week; 1-3 meals/month; 4-10 meals/year; 1-3 meals/year. This is the reasoning we used in the present study.

·          The use of figures was excellent, they were beautiful and attention grabbing, and they really helped me to understand the data you presented.  Make certain that legends are available for the colors and that the colors are clearly different to improve these a little bit and reduce confusion.

Answer: Figures have been improved and now the Y axes are added together with the designations.

Abstract: 

·          Mention that Chukotka is in the Russian Arctic and home to indigenous people in the abstract.

Answer: We are currently revising the third article in the series “Traditional Diet and Environmental Contaminants in Coastal Chukotka” of four articles. The first one is “Study design and dietary pattern” (already accepted by the journal for the publication), the second one is “Legacy POPs”, the fourth one is “Recommended Intake criteria”. In the previous articles we have already talked about Russian Arctic and indigenous people.

·          Spell out the minerals of interest the first time they are used and put the chemical symbols in parentheses (Lead (Pb) and mercury (Hg)).  The mineral name should always be spelled out if it is the first word of a sentence. 

 Answer: all the metals symbols have been defined in the text the first time they are used. Also all the names of metals are spelled out if it is the first word of a sentence.

·          The term “concentrations” is preferred to the word “levels” to improve clarity.

 Answer: In our view the words “concentration” and “level” can be used interchangeably in this context, and we choose to use both in order to avoid the constant using the word “concentrations,” which has many instances of use in the text. We have substituted the word “levels” by the word “concentrations” in some places (subsections names, table’s names). We have left the use of “levels” in Russian Allowable Levels, because it is the literal translation of the title.

·          “ascidians (sea squirts) contained Al . . . “

 Answer: The words “(sea squirts) contained” have been inserted in the Abstract.

·           “Follow-up over 15 years of . . . “

 Answer: the sentence has been improved to say: “Follow-up (15 years after the first one) of Hg, Pb, Cd concentrations in local foods has not revealed any increase …).

·          Introduction:  The introduction is very long and poorly referenced.  Additionally, some of the information mentioned here is repeated later in the discussion.

 Answer: In our view the length of the Introduction is appropriate and effective for introducing the work. We added some more references and reorganized the material to some extent.

First paragraph:

·          “Metals are persistent toxic substances (PTS) . . . and do not undergo degradation”  Elements can undergo certain chemical changes, but they don’t undergo degradation under normal conditions on earth.

 Answer: the sentence has been improved: the phrase “do not undergo” has been inserted.

·           “Characteristics of some metals include global transportation, often with Arctic deposition, accumulation and transport in rivers, seas, and oceans; bioaccumulation and biomagnification, especially in marine food webs; slow elimination from organisms, including humans; and variable toxicity which can include adverse human health effects.”  The metals you tested for are very variable and don’t all have the same characteristics.  Additionally, while one could argue that they can be metabolized to different elemental forms (for example, methylation of metallic mercury by sedimentary microbes) it’s misleading to talk about metabolism of elements, so I changed this to “elimination” which is more clearly definable in this case.  It should also be noted that some of the metals you are looking at are nutrients required for health and belong in the food chain at specific concentrations.

 Answer: the sentence has been improved.

·          “Some metals cross the placental barrier, where they can affect the fetus during the development, and can be excreted in milk.”

 Answer: the sentence has been improved.

·          Second paragraph:  “have accumulated and cycled in the environment . . . “

 Answer: the sentence has been improved.

·          “significant quantitiets of metals are now introduced and redistrubted in the environment from human activities at local, regional, and global scales, including fossil fuel combustion. . . transportation, and waste disposal.

 Answer: in our opinion it is better to leave it as it is.

·          Third paragraph:  references are needed for the first, second, and third sentences.

 Answer: references have been provided.

·          Fourth paragraph:  This paragraph is also missing references for the information presented.

 Answer: references have been provided.

·          What does “different intensities” mean in the second sentences? 

 Answer: this phrase was deleted.

·           “Seaweed, benthic molluscks (bivalves, eg mussels, and gastropods) . . . “Note that ascidians are sea squirts here and in the abstract. 

 Answer: corrections have been done.

·           “lichen ‘saturated’ with metals” what do you mean here by “saturated”?  Would it be more clear to say “lichen contaminated with high concentrations of metals?” 

 Answer: In our opinion the term “saturated’ is acceptable here; it reflects the high sorption capacity of lichens.

·           “and can be used as indicators of industrial pollution.”  (omit “of the territories”)

 Answer: this phrase was deleted.

·          . . . “Natives as foods, also have the ability to accumulate metals from the soil.”

 Answer: corrections have been done.

·          Sixth paragraph:  Why is meta-analysis in quotation marks?  It should say “A meta-analysis was applied in 83 . . .  in Arctic biota with the purpose determining if temporal trends exist in the data collected over the past decades.”  Can you include dates during which the data was collected here?

  Answer: the sentence was improved; dates were inserted.

M&M

·          Field sampling:  “Samples of fish (marine, migratory, freshwater), meat of terrestrial mammals (reindeer, hare), meat and blubber of marine mammals . . . “

 Answer: the “mammals” was relocated.

·          A map of the area of the settlements and other collection areas would be helpful.

 Answer: the map is presented in the article 1 (Study Design and Dietary Pattern) of the present series of 4 articles.

·          Was defined earlier in this manuscript?  You need to spell out the name of this organization once.

 Answer: NWPHRC is defined now.

·          Were any bird samples collected besides the goose?  Were any mammalian liver and kidney samples collected?  This is unclear.  If they were not collected, should they be mentioned?

Answer: No additional bird samples were collected. Liver and kidney were not available at the time of our fieldwork and were absent in the storages of local people. We do have to emphasize that although liver and kidney are generally strong accumulators of some metals and should be regarded as additional sources of total dietary exposure to metals, however the amounts of liver and kidney consumed by local Chukotka people are insignificant.

·          Second paragraph:  A sentence should not start with a number.  “Samples from multiple specimens of each biological species similar in age and size were pooled.  Pooled samples accounted for 37% of the total sample number.”

 Answer: corrections have been done.

·          Spelling error in next sentence: “pieces of approximately equal . . .”  How did you determine that pieces were the same size?  Were they weighed to be certain? 

Answer: this phrase was deleted.

·          What size is a “medium sized cap” from a mushroom?  How much does a medium sized cap weigh? It would be good if you had precise measurements of the samples that were pooled. 

Answer: this phrase was deleted.

·          Metals Analyzed section:  This can be combined with the “Chemical Methods” section.

 Answer: in our opinion it is better to leave it as it is.

·          Chemical Methods section:  POPs are persistent organic pollutants, which you are not studying.  Why are they mentioned in the title of this section?

 Answer: POPs were deleted. It is the echo of the Article 2 on POPs.

·          Define RPA.

Answer: defined.

·          Define AMAP.

Answer: defined.

·          Second paragraph:  I am not familiar with the term “mineralized” being used this way.  We usually use the term “digested” for this process.  Mineralized suggests you are adding mineral to the sample.  Were samples for mercury analysis digested in the same way as samples for the other metals?

Answer: corrected.

·          Use subscripts appropriately in chemical formulas, or spell out the reagent:  HNO3 = nitric acid.

Answer: corrected.

·          MARS-5 microwave digester.

Answer: corrected.

·          LOD needs to be spelled out (limit of detection?)  Also, an abbreviation should not be used as the first word of a sentence.

Answer: corrected.

Processing, analysis:

·          “Metal concentrations from coastal Chukotka local foods were compared between species, and species from coastal Chukota were compared to the same species in other circumpolar regions to assess temporal trends.” 

 Answer: in our opinion it is better to leave it as it is, but with substitution of the “same” by the “corresponding”.

Results: 

·          Figure 1:  “Concentrations of metals. . . “ The blue colors for mushrooms and wild plants are too similar to tell apart.  Additionally, you can’t see the mean and median markers within the bars.

Answer: Figure was improved, the blue colors for mushrooms have been changed pink; bars have been adjusted to become more visible.

·          Levels of metals:  Should be “concentrations” of metals.

Answer: corrected.

·          Sentences should not start with a chemical symbol.  The element name must be spelled out, or the sentence reconstructed: “All foods were very low in Pb”

Answer: corrected.

·          “Seafood contained the highest concentrations of Cd”. Avoid using dashed lines between the element and information. 

Answer: correction has been done, but using another dash. Here the dashed lines allow escaping from the constant repeat of the phrase “highest concentrations”.

·          Table 2. “The highest concentrations of metals in seafood samples . . . “

Answer: corrected.

·          Figure 2:  “Concentrations of As . . . “

Answer: corrected.

·          Follow-up:  “Although there was a lack of variation in concentrations of the majority of metals, there was an obvious tendency for declining metal concentrations in arctic char . . . “

Answer: corrected.

·          Table 3:  You need to mention that you are comparing Chukotka and neighboring Chukotsky.

 Answer: in our opinion it is better to leave it as it is, because the explanation that we are comparing the neighboring districts of coastal Chukotka is provided right before the Table 3.

Estimated daily intake:

·          You already explained what EDI was above, so you can omit the parenthesis.  “Questionnaires were assessed to determine EDIs of metals based on local food consumption and average concentrations of metals . . . “

 Answer: the first paragraph has been removed to the section “Materials and methods” and was inserted instead of the former sentence about EDIs.

·          Does it make sense to estimate 150 g of each foodstuff per meal per day?  I am trying to understand how this worked:  if someone said they consumed whale blubber and mushrooms, you assumed 150 g of each?  This seems both arbitrary and unlikely to be accurate.

 Answer:  1) It was impossible within our research setting to administer a personalized 24-hour dietary recall survey. We therefore conducted a broad-based survey on dietary consumption practices, using standard questionnaire. Standard questionnaires are known to be less effective in reflecting the actual consumption practice (particularly in the Arctic), but that was the best available option. 2) We are not nutritionists; we are environmental health researchers. Our main task is to assess the dietary food exposure (to contaminants) of a community, based on the values of local food contamination. 3) The principle to which we adhere over the many years of research in the Russian Arctic is as follows: a thorough assessment of the frequencies of each food item intake is much more informative and reliable as a criterion of real food intakes than an estimation of a portion size; 4) individual fluctuations of the self-reported portion size vary substantially and we need to be able to average these numbers per day/week/month/year; 5) Given that the assumed rage of 100g to 200g is used for meat and fish portions (as well as side dish or salad) around the world, it is easier and more reliable to use conventionally assumed one portion size as 150g/meal of each foodstuff, and then to average the reported frequencies: 1-3 meals/day; 4-6 meals/week; 1-3 meals/week; 1-3 meals/month; 4-10 meals/year; 1-3 meals/year. This is the reasoning we used in the present study.

·          Second paragraph:  You mention POPs but I don’t think they were defined before. POP was also used previously in a methods subsection header where it didn’t make sense.

 Answer: POPs are covered in the previous Article 2 of this series.

·          “. . . seafoods are extremely powerful carriers of metals, with the following EDIs:”

 Answer: “of metals, with the following” was inserted in the text.

·          Why do you have “makes” in quotation marks after fowl?  “Fowl in the diet contributes 50% of the Pb . . .

 Answer: corrected.

·          Remove quotation marks from “contributors”

 Answer: corrected.

·          “Berries are a substantial source of Mn. . . “

  Answer: corrected.

·          Figure 3: The pie charts are really helpful and attention-grabbing. You need to have a legend to explain the different colors because some areas of the pie chart are too narrow to read.

 Answer: in our opinion it is important to show the main food groups responsible for the intake of metals. Almost all colors of a pie (which contribute more than 5%) are defined by the inscriptions. A legend containing all foods for the pie chart will be very massive and therefore the colored definitions will be very small and hardly visible.

Hygienic regulations:  

·         
Allowable levels doesn’t need to be capitalized.  Four should be spelled out.

  Answer: the capitalized “Allowable Levels” is the official name of the Russian document; it is better to use this cliche. Four is spelled out.

·          “which is advantageous for food contamination assessment, as the Codex Alimentarius (13) has fewer standards.”

Answer: corrected.

·          “Standards for several other metals . . . were established in the 1980s in several country foods by the USSR and have not been revisited. “

 Answer: corrected.

·          Table 4:  Do you need to include liver and kidney if they were not collected in this study?

 Answer: in our opinion it is very useful for the readers to have a chance to compare the standards for different tissues. Also it is important to show that the standards for raw viscera are available at least in Russia (and in former USSR).

Exceedances of analyzed concentrations:

·          Second paragraph:  “Exceedances over the allowable levels of Cr are observed . . . “

Answer: corrected.

·          Drinking water:  you can use Cd for cadmium throughout this paragraph.

Answer: corrected.

·          “ingest” doesn’t need to be in quotation marks.

Answer: corrected.

·          Discussion:  This section needs a little reorganization so that you incorporate your data into the discussion better.  Compare your findings directly to findings from other studies, rather than listing the findings from the other studies and then summarizing your results for those elements at the very end of the section. 

Answer: In our opinion the organization of this section is effective and understandable for readers.

·          Don’t begin a sentence with an abbreviation, but you can use the chemical symbols for elements elsewhere throughout this section.

Answer: corrected.

·          Genus and species names should be italicized. 

Answer: corrected.

·          Table 6:  Concentrations of selected metals in meat, blubber, and mantak sampels of grey whale from Chukotka (present study) and bowhead whale from Alaska . . .”

Answer: corrected.

·          Conclusions:  Can you say that the study is comprehensive if you’ve already noted that you don’t have enough samples for statistical power?

Answer: Agree. We put the following sentence: “To our knowledge this is the only diversified study on multiple metals in the variety of local subsistence foods from the coastal Chukotka since the beginning of 2000s”.

Third paragraph:

·          First sentence:  “. . . has revealed the following exceedances over the Russian Allowable Levels: “  The source of the exceedance does not need to be in parenthesis.  You can separate the different groups of elements and foodstuffs using semicolons instead of commas to reduce confusion here.

Answer: corrected.

Reviewer 2 Report

Comments

Title: Traditional Diet and Environmental Contaminants in Coastal Chukotka III: Metals

No. ijerph-400195

A paper sent for review entitled " Traditional Diet and Environmental Contaminants in Coastal Chukotka III: Metals" concerns the important problem of pollution of 18 metal(loid)s elements (i.e., As, Zn, Cd, Pb and Mn, etc) in terrestrial, freshwater and marine biota samples collected from coastal Chukotka and health risk resulting from the current pollution.

Due to not well-planned studies, Authors have not fully complied with generally accepted requirements that are put into scientific works published in international journals.

Overall, there were a number of grammatical issues within the manuscript. I suggest the authors hire a copyeditor to correct these syntax issues. In general, I am not convinced by the English language and I think that the manuscript needs revisions.

Some specific comments:

1. The title of MS is not adequate.

2. Figures and tables:

I think they are all in poor organization and have low readability. The caption of X and Y axes? The figures all not clear.

Table 4 is not necessary. This is not the results of this study. It should be deleted.

3. I am worried about the reliability of data, because of numbers of samples are too little. For example, only one sample of Grayling, or Chum salmon and so on was collected and analysis. It is not representative of the actual pollution, due to a certain degree of heterogeneity in coastal area. 

Author Response

Comments and Suggestions for Authors

Comments

Title: Traditional Diet and Environmental Contaminants in Coastal Chukotka III: Metals

No. ijerph-400195

A paper sent for review entitled " Traditional Diet and Environmental Contaminants in Coastal Chukotka III: Metals" concerns the important problem of pollution of 18 metal(loid)s elements (i.e., As, Zn, Cd, Pb and Mn, etc) in terrestrial, freshwater and marine biota samples collected from coastal Chukotka and health risk resulting from the current pollution.

·         Due to not well-planned studies, Authors have not fully complied with generally accepted requirements that are put into scientific works published in international journals.

Answer: we have sufficient experience of publishing in international journals, and we strictly followed the requirements of preparation the articles for the IJERPH.

·         Overall, there were a number of grammatical issues within the manuscript. I suggest the authors hire a copyeditor to correct these syntax issues. In general, I am not convinced by the English language and I think that the manuscript needs revisions.

Answer: we did our best in improving the grammar.

Some specific comments:

·         The title of MS is not adequate.

Answer: We are currently revising this third article, which is part of the series “Traditional Diet and Environmental Contaminants in Coastal Chukotka” of four articles. The first one is “Study design and dietary pattern” (already accepted by the journal for the publication), the second one is “Legacy POPs”; the fourth one is “Recommended Intake criteria”. All the titles are very concrete and understandable.

·         Figures and tables: I think they are all in poor organization and have low readability. The caption of X and Y axes? The figures all not clear.

Answer: Figures have been improved and now the Y axes are added together with the designations.

·         Table 4 is not necessary. This is not the results of this study. It should be deleted.

Answer: Table 4 should be very useful for readers, particularly non-Russians, because the international standards regarding metals are rather poor in the country foods. Data from Table 4 make it easy for readers to compare the different national standards, or to assess the levels of contamination of foods by metals in the countries where the standards for some metals are lacking. 

·         I am worried about the reliability of data, because of numbers of samples are too little. For example, only one sample of Grayling, or Chum salmon and so on was collected and analysis. It is not representative of the actual pollution, due to a certain degree of heterogeneity in coastal area. 

Answer: Our study settlements are remote communities that are difficult to reach due to extremely limited transportation options that are also vulnerable to weather. We worked within the logistical and budgetary constraints to collect samples of the key species (as inclusive and diverse as possible) of local fauna and flora to assess the present day contamination of the maximum wide spectrum of species, which are consumed by local people, being the main sources of exposure to metals, and to compare these detected values with previous studies in the region and in neighboring foreign Arctic territories. Under such circumstances each sample (collected and analyzed) is very important for the entire goal and different tasks of the project.

Reviewer 3 Report

In this study the Authors performed a study to measure 18 environmental metals in samples of locally harvested terrestrial, drinking water, freshwater and marine biota collected in 2016 in coastal Chukotka

Even if the topic is very interesting, it's lacking in different points:

1.      The Introduction focuses mainly on the background of studies about metals in local food and in the different biota species. On the contrary the Authors do not discuss their health effects. The introduction should be improved by adding informations about it (i.e. consider “Toxicity, mechanism and health effects of some heavy metals doi: 10.2478/intox-2014-0009” and “Non-occupational exposure to heavy metals of the residents of an industrial area and biomonitoring doi:10.1007/s10661-016-5693-5").

2.      In Introduction the Authors should discuss more in depth metals in drinking water and in Matherials and Methods more informations should be provided about the types of drinking water that have been collected.

3.      In Matherials and Methods the Authors should be more specific about the questionnaire (i.e.  is there a questionnaire validation protocol? how many subjects responed to the questionnaire)

4.      The Authors should point out the limits of the study in Discussion Section (i.e. why was not biological monitoring done?).

5.      Some of the metals, like arsenic and chromium, also have carcinogenic effects. The authors should emphasize this aspect in Introduction or in Discussion (i.e. consider  “IARC Chromium (VI) compounds. International Agency for Research on Cancer monographs on the evaluation of carcinogenic risks to humans”, “IARC  Monogr Eval Carcinog Risk Chem Hum Arsenic, metals, fibers and dusts. Volume 100C” and “ “Environmental exposure to arsenic and chromium in an industrial area doi: 10.1007/s11356-017-8827-6").

Author Response

Comments and Suggestions for Authors

In this study the Authors performed a study to measure 18 environmental metals in samples of locally harvested terrestrial, drinking water, freshwater and marine biota collected in 2016 in coastal Chukotka

Even if the topic is very interesting, it's lacking in different points:

1.      The Introduction focuses mainly on the background of studies about metals in local food and in the different biota species. On the contrary the Authors do not discuss their health effects. The introduction should be improved by adding informations about it (i.e. consider “Toxicity, mechanism and health effects of some heavy metals doi: 10.2478/intox-2014-0009” and “Non-occupational exposure to heavy metals of the residents of an industrial area and biomonitoring doi:10.1007/s10661-016-5693-5").

Answer: Health effects and toxicological aspects of exposure to metals are out of the scope of the present article. This series of articles are aimed at the assessment of traditional diet of indigenous people of coastal Chukotka, their dietary exposure through the local food webs and setting the Recommended Food Daily Intake Limits (RFDILs) for the studied food items.

2.      In Introduction the Authors should discuss more in depth metals in drinking water and in Matherials and Methods more informations should be provided about the types of drinking water that have been collected.

Answer: 2.1. With regard to the Introduction section - drinking water in the studied settlements of Providensky district was assessed in the present study for the first time. The literature on metals in drinking water in Chukotka region generally and in Providensky district in particular is entirely absent.

With regard to the Matherials and Methods section - the following information is provided:  “All drinking water samples have been collected in each settlement, as well as in Provideniya and Anadyr (for comparison needs); each 500 ml water sample was put into a plastic container with a screw top”.

Answer: 2.2. We are currently revising this third article, which is part of the series of four. In the first article “Study design and dietary pattern” (already accepted by the journal for publication) we explain that “the municipal water delivery in the study settlements is facilitated through scheduled service by a water truck. Untreated water, pumped from nearby rivers and lakes, is typically unloaded into large barrels (200–250 liters), kept inside a residence, using a hose”.

3.      In Matherials and Methods the Authors should be more specific about the questionnaire (i.e.  is there a questionnaire validation protocol? how many subjects responed to the questionnaire)

Answer: see the answer 2.2. The detailed information on the questionnaire aspects is provided in the first article.

4.      The Authors should point out the limits of the study in Discussion Section (i.e. why was not biological monitoring done?).

Answer: The present study on environmental metals in Providensky district of Chukotka is the first ever study of this sort carried out in this region. As for the entire coastal Chukotka, the neighboring Chukotsky district was the only one where the collection and chemical analysis of local foods for Hg, Cd and Pb (only 3 metals) were conducted in 2001-2002. Monitoring has never been conducted in the studied regions.

5.      Some of the metals, like arsenic and chromium, also have carcinogenic effects. The authors should emphasize this aspect in Introduction or in Discussion (i.e. consider  “IARC Chromium (VI) compounds. International Agency for Research on Cancer monographs on the evaluation of carcinogenic risks to humans”, “IARC  Monogr Eval Carcinog Risk Chem Hum Arsenic, metals, fibers and dusts. Volume 100C” and “ “Environmental exposure to arsenic and chromium in an industrial area doi: 10.1007/s11356-017-8827-6").

Answer: Health effects and toxicological aspects of exposure to metals are beyond the scope of the present article.

Round 2

Reviewer 2 Report

 All the figures in revised file still looks odd. I don't think the figures was improved a lot. 

X and Y-axis labels should be added in all figures.

Author Response

Answer: In the first round I have already submitted (together with the revised manuscript) the improved versions of the Figures 1 and 2 - axes definitions were provided and the color of mushrooms boxes in the Figure 1 has been changed from blue to pink. Two separate gif files have been submitted, but the figures have not been inserted into the text of the manuscript.

Now the improved figures are inserted.

Reviewer 3 Report

The Authors revised the article on the basis of the observations that have been made, however all references to the effects of metal toxicity on human health are still lacking. It is essential to underline this aspect because some metals are also carcinogens.

This is a serious lack because every effort made by international scientific organizations to establish limits and reference values is related to the protection of human health. 

Author Response

Answer: according to your suggestion we added short information on carcinogenicity of metals to the section Introduction, and made a link to the IARC classification.

We would not like to go in deep on the toxicological aspects and health effects issues of exposure to metals because this term is beyond the scope of the present article.

In the fourth article (of the present series of four) which is dealing with the Recommended Food Daily Intake Limits (RFDILs) for the studied food items, we are not discussing the carcinogenicity of the studied metals. We use the established values of the oral Tolerable Daily Intakes (TDIs) for inorganic metals which are all Non-Carcinogenic Toxicological Reference Values (TRVs) Recommended for Use in Human Health Risk Assessments.

The only exception is inorganic arsenic for which the lower limit on the benchmark dose for a 0.5% increased incidence of lung cancer (BMDL0.5) was determined from epidemiological studies to be 3.0 μg/kg bw per day (2–7 μg/kg bw per day based on the range of estimated total dietary exposure) using a range of assumptions to estimate total dietary exposure to inorganic arsenic from drinking-water and food (WHO TRS 959-JECFA 72. Evaluation of certain food additives and contaminants. 72 Report of the Joint FAO/WHO Expert Committee on Food Additives. World Health Organization. Technical Report Series 959. Geneva, 2011).